# Dimensional crossover in semiconductor nanostructures

Matthew P. McDonald[1], Rusha Chatterjee[1], Jixin Si[2], Boldizsár Jankó[2] & Masaru Kuno[1]

Recent advances in semiconductor nanostructure syntheses provide unprecedented control over electronic quantum confinement and have led to extensive investigations of their size- and shape-dependent optical/electrical properties. Notably, spectroscopic measurements show that optical bandgaps of one-dimensional CdSe nanowires are substantially (approximately 100 meV) lower than their zero-dimensional counterparts for equivalent diameters spanning 5–10 nm. But what, exactly, dictates the dimensional crossover of a semiconductor's electronic structure? Here we probe the one-dimensional to zero-dimensional transition of CdSe using single nanowire/nanorod absorption spectroscopy. We find that carrier electrostatic interactions play a fundamental role in establishing dimensional crossover. Moreover, the critical length at which this transition occurs is governed by the aspect ratio-dependent interplay between carrier confinement and dielectric contrast/confinement energies.

[1] Department of Chemistry and Biochemistry, University of Notre Dame, Notre Dame, Indiana 46556, USA. [2] Department of Physics, University of Notre Dame, Notre Dame, Indiana 46556, USA. Correspondence and requests for materials should be addressed to M.K. (email: mkuno@nd.edu).

ntuitive expectations, based on a framework of non-interacting particles, suggest that reducing the length of a nanoscale system's confining potential gradually induces the emergence of quantum confinement[1–3] effects in its spectroscopic response. This crossover is expected to be smooth, without the appearance of a critical length. However, the presence of interactions fundamentally alters this picture of dimensional crossover in many-particle systems. Several low-dimensional systems, in fact, show interaction-induced phases of matter absent in higher dimensions. They include fractional quantum Hall states in two dimensions[4], Tomonaga–Luttinger liquids in one-dimension (1D)[5] and Kondo effects in zero-dimensional (0D) quantum dots (QDs)[6]. Here we demonstrate that analogous many-particle interactions dictate the 1D-to-0D dimensional crossover in low-dimensional semiconductors (Fig. 1). Our spectroscopic measurements and supporting theoretical calculations indicate that this transition occurs at a critical length determined by the delicate balance between carrier confinement and electrostatic interaction energies.

While past studies have attempted to investigate $\sim 100$ meV bandgap ($E_g$) differences between 1D and 0D CdSe nanostructures by probing nanorods (NRs) of controlled length[7–9] no consensus exists as to when a 1D object exhibits 0D character[7,9]. Although general trends have been gleaned, they are largely the result of ensemble measurements, which suffer from inhomogeneous broadening due to inherent size-and-shape distributions[10–12]. Importantly, they rely on photoluminescence and tunnelling spectroscopies[7,9]. However, dark/bright exciton splitting[13], trap-induced Stokes shifts[14] and enhanced exciton binding energies[15] can significantly alter perceived bandgaps, complicating accurate $E_g$ estimates.

Here we directly probe this 1D-to-0D transition using single nanowire (NW)/NR absorption spectroscopy to eliminate ambiguities as to the actual evolution of electronic structure across dimensionality. This is accomplished using spatial modulation microscopy[12,14,16,17], which entails modulating a NW/NR's position in and out of a focused laser beam. Subsequent lock-in detection measures the transferred transmitted laser power modulation, which relates to individual nanostructure extinction cross-sections (Supplementary Note 1 and Supplementary Fig. 1)[17]. As obtained experimental data are found to be in good agreement with an effective mass model that explicitly includes dielectric contrast/confinement effects, and, consequently, provides a critical length at which 1D nanostructures become 0D.

## Results

**Spectroscopic observations.** Figure 2a shows the absorption spectrum of an individual CdSe NW from a diameter ($d$) $6.8 \pm 1.2$ nm ($b \geq 5$ μm) ensemble. Figure 2b,c illustrate corresponding absorption spectra of single CdSe NRs from $d = 6.7 \pm 1.1$ nm (length; $b = 160 \pm 55$ nm) and $d = 6.8 \pm 0.7$ nm ($b = 30.4 \pm 2.6$ nm) ensembles (Supplementary Note 2 and Supplementary Fig. 2). Three to four transitions (labelled α, β, γ, and δ) are apparent in each spectrum and are excitonic in nature as predicted by a model which explicitly accounts for both spatial confinement and enhanced electrostatic interactions in NRs[15]. These states can be explicitly linked to analogous α, β, γ, and δ transitions in individual CdSe NWs[12,14,16]. The data in Fig. 2 therefore represent the first direct measurements of single CdSe NR absorption spectra.

Most notable is an overall $\sim 30$ meV average α blueshift in $b \sim 30$ nm NRs (Fig. 2c) relative to those of longer, equi-diameter particles (Fig. 2a,b) ($b \sim 30$ nm: α = $1.904 \pm 0.025$ eV; $b \sim 160$ nm, α = $1.872 \pm 0.016$ eV; NW: α = $1.874 \pm 0.022$ eV). In fact, probing 25 individual $b \sim 30$ nm NRs reveals α blueshifts up to 64 meV. Figure 3 illustrates this, showing absorption spectra of three different $b \sim 30$ nm NRs.

The simplest explanation for this behaviour stems from an increase in electronic confinement as NR lengths decrease. In particular, carriers experience additional confinement along the NR $z$-axis, adding to the radial ($\rho$) confinement present exclusively in NWs[12,15,16]. To assess such confinement effects, we have constructed a modified version of an effective mass model previously used to explain absorption spectra of individual CdSe NWs[12,14,16].

**Effective mass model.** In the model, NW/NR electron wavefunctions are given by

$$\Psi_{e, \pm 1/2}^{(n, n_z, m)} = \frac{u_{\pm 1/2}}{a\sqrt{\frac{\pi b}{2}} J'_m(\alpha_{n,m})} J_m\left(\frac{\alpha_{n,m}}{a}\rho\right) \sin\left(\frac{n_z \pi}{b}z\right) e^{im\phi}. \quad (1)$$

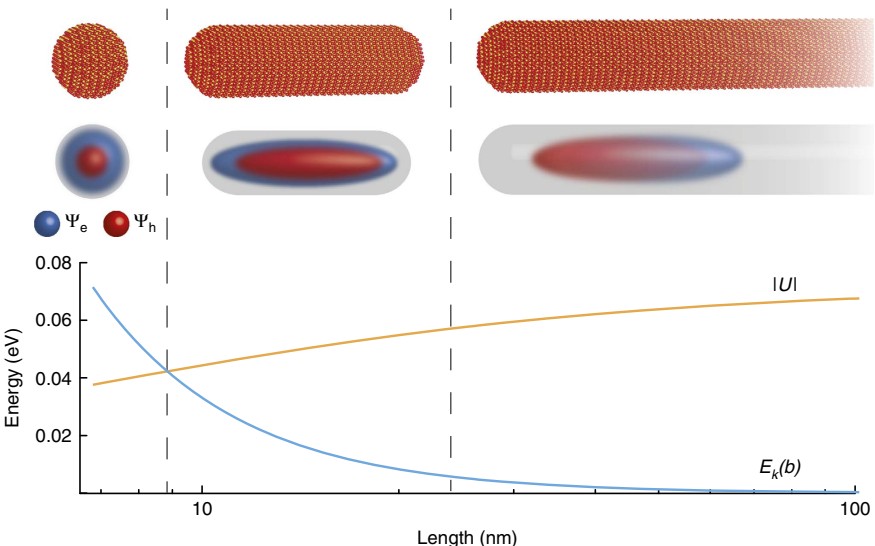

**Figure 1 | One-dimensional to zero-dimensional crossover of a semiconductor nanowire's electronic structure.** Top row: structural evolution of a nanowire into a quantum dot. Middle row: corresponding evolution of nanowire and nanorod electron and hole wavefunctions. Bottom row: a plot depicting the interplay between aspect ratio-dependent carrier confinement, $E_k(b)$ and dielectric contrast/dielectric confinement electrostatic energies, $|U|$.

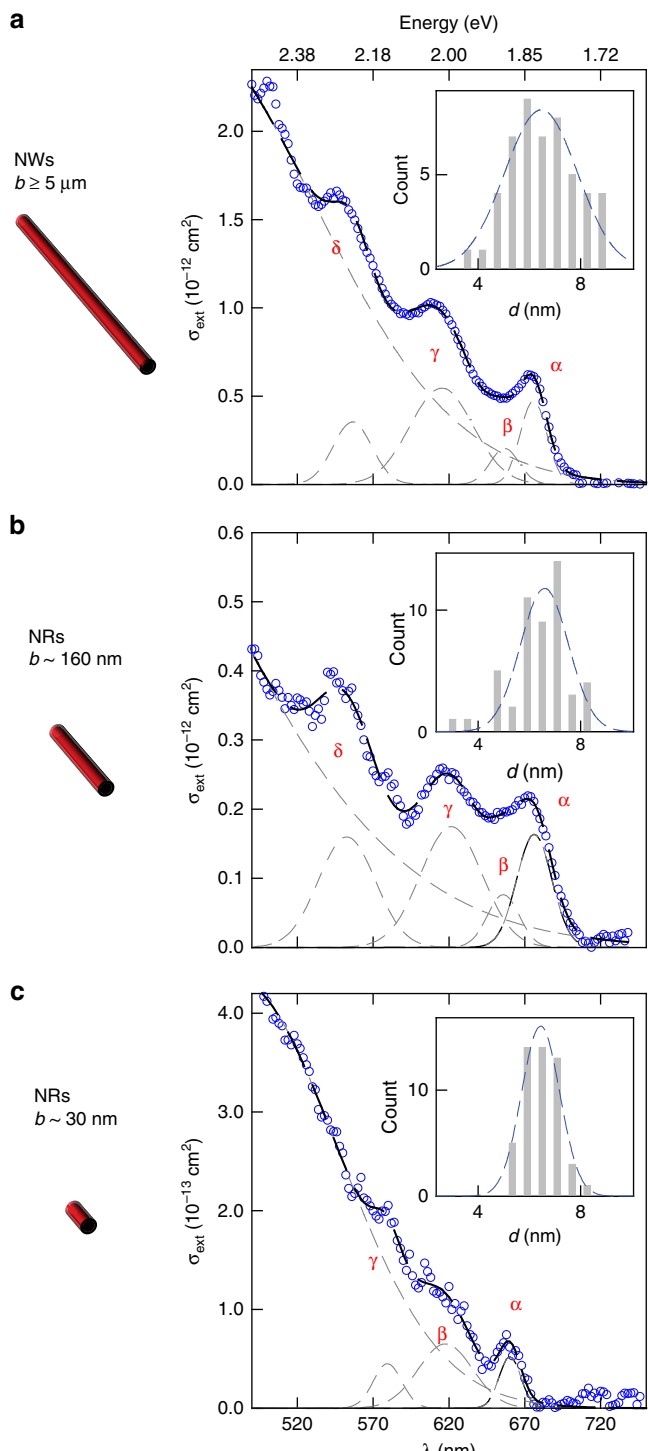

**Figure 2 | Absorption spectra of individual CdSe nanowires and nanorods.** Absorption spectrum of an individual (**a**) nanowire from a $d = 6.8 \pm 1.2$ nm ($b \geq 5\,\mu$m) ensemble; (**b**) nanorod from a $d = 6.7 \pm 1.1$ nm ($b = 160 \pm 55$ nm) ensemble; and (**c**) nanorod from a $d = 6.8 \pm 0.7$ nm ($b = 30.4 \pm 2.6$ nm) ensemble. Blue open symbols represent measured extinction values plotted as a function of wavelength ($\lambda$). Corresponding peak $\alpha$ extinction cross-sections are $\sigma_{ext} \sim 7 \times 10^{-13}$ cm$^2$ ($b \geq 5\,\mu$m), $\sigma_{ext} \sim 2 \times 10^{-13}$ cm$^2$ ($b \sim 160$ nm), and $\sigma_{ext} \sim 8 \times 10^{-14}$ cm$^2$ ($b \sim 30$ nm). Spectra are fit to a sum of Gaussians (black dashed line) from where individual transitions (grey dashed lines) are extracted. Each sample's sizing histogram is inset in (**a–c**). S.d. reported.

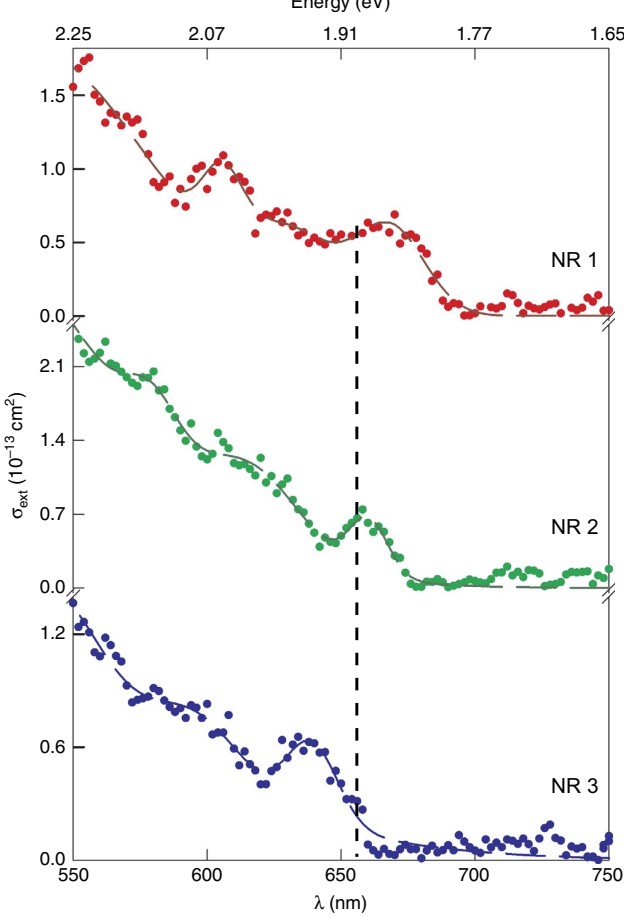

**Figure 3 | Absorption spectra of individual CdSe nanorods.** Three individual CdSe nanorod absorption spectra obtained from the same ensemble ($d = 6.8 \pm 0.7$ nm, $b = 30.4 \pm 2.6$ nm). The solid (red, green and blue) circles are experimental data points with corresponding (red, green and blue) dashed lines as their sum of Gaussians fit. The dashed vertical line represents the average $\alpha$ energy of 25 individual $b \sim 30$ nm rods.

$u_{\pm 1/2}$ is the electron Bloch function, $a$ is the NW/NR radius, $b$ is the corresponding length, $J_m(x)$ are Bessel functions of the first kind and $\alpha_{n,m}$ is the $n$th root of the $m$th order Bessel function. Parameters ($n$, $n_z$, $m$) are radial, longitudinal and angular quantum numbers, respectively. Hole wavefunctions are linear combinations of effective heavy-hole ($|HH\rangle_{1(2)}$) and light-hole ($|LH\rangle_{1(2)}$) states, given by[18]

$$\Psi_h^{(F_z, n_z)} = \mathcal{A}|HH\rangle_1 + \mathcal{B}|HH\rangle_2 + \mathcal{C}|LH\rangle_1 + \mathcal{D}|LH\rangle_2 \quad (2)$$

with $\mathcal{A}$, $\mathcal{B}$, $\mathcal{C}$ and $\mathcal{D}$ their relative weights; $F_z$ is the angular momentum projection onto the NW/NR $z$-axis. Importantly, longitudinal kinetic energy terms are not assumed to be negligible in comparison to radial confinement (that is, $k_z \neq 0$)[9,15,18]. Full expressions and derivations can be found in Supplementary Note 3.

Corresponding quantum size level energies for $b \sim 30$ nm NRs only increase $\sim 3$ meV over the $k_z = 0$ case, consistent with previous modelling[9] (Supplementary Fig. 3). Clearly, simply accounting for longitudinal carrier confinement cannot explain the $\sim 30$ meV average blueshift seen in Figs 2 and 3. Additionally, even though Supplementary Fig. 4 shows that obtained $b \sim 30$ nm NR spectra are blueshifted ($\sim 18$ meV) relative to $\alpha$ of the corresponding ensemble spectrum, the residual $\sim 12$ meV blueshift cannot be explained via confinement alone.

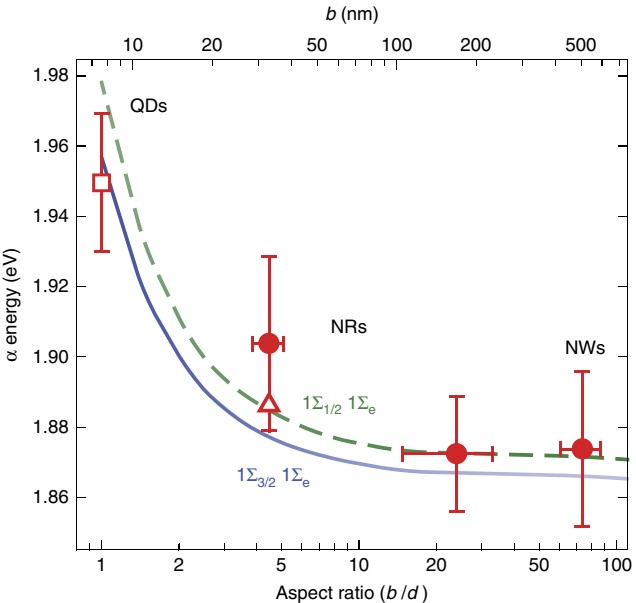

**Figure 4 | Evolution of CdSe nanoparticle optical properties across dimensionality.** Extracted (average) α energies plotted as a function of aspect ratio ($b/d$) for all three wire/rod samples (solid red circles) as well as tabulated quantum dot literature values (open square) ($d = 6.8$ nm)[2,22,23]. S.D. reported. The open red triangle is the $b \sim 30$ nm ensemble spectrum α energy. Superimposed over the data are theory lines for the first ($1\Sigma_{3/2}1\Sigma_e$; solid blue line) and second ($1\Sigma_{1/2}1\Sigma_e$; dashed green line) one-dimensional excitons[15]. Relative transition strengths for each are indicated by the transparency of the lines. Average nanowire and $b \sim 160$ nm nanorod α energies were obtained using weighted individual wire/rod energies with weighting factors obtained from a literature-compiled sizing curve and transmission electron microscopy-derived diameter distributions (Supplementary Note 5, Supplementary Note 6 and Supplementary Fig. 2).

**Electrostatic contributions.** A more complete explanation must therefore consider the dimensional evolution of carrier electrostatic effects wherein two contributions exist. The first is dielectric contrast, which stems from dielectric constant ($\varepsilon$) differences between a nanoparticle and its immediate surroundings[12,15,19]. The second is dielectric confinement, which arises due to repulsive 'mirror' forces at the particle/medium dielectric interface[12,15]. In 1D systems, dielectric contrast outweighs dielectric confinement, lowers α's predicted energy by $\sim 60$ meV and leads to the formation of 1D-excitons in CdSe NWs[12,15,16,20]. In QDs, electrons/holes effectively screen each other at every point such that their electrostatic contributions to overall carrier energies are vanishingly small[15,21]. Aspect ratio-dependent electrostatic effects therefore rationalize average α shifts observed in Figs 2 and 3.

To explicitly account for how CdSe's electronic structure transitions from 1D-to-0D, we model these electrostatic effects as functions of NW/NR aspect ratio ($b/d$). In practice, this entails solving Poisson's equation inside a finite length ($b$) dielectric cylinder to find the potential ($V(\mathbf{r}, \mathbf{r}_0)$) at an arbitrary position $\mathbf{r}$ due to a point charge at $\mathbf{r}_0$. The corresponding electrostatic energy is given by[15]

$$U(\mathbf{r}_e, \mathbf{r}_h) = -\frac{q^2}{4\pi\varepsilon\varepsilon_0|\mathbf{r}_e - \mathbf{r}_h|} - qV(\mathbf{r}_e, \mathbf{r}_h) + \frac{q}{2}V(\mathbf{r}_e, \mathbf{r}_e) + \frac{q}{2}V(\mathbf{r}_h, \mathbf{r}_h)$$

(3)

where $\mathbf{r}_e = (\rho_e, \phi_e, z_e)$ ($\mathbf{r}_h = (\rho_h, \phi_h, z_h)$) is the electron (hole) position, and $q$ is the elementary charge. The first two terms in

equation (3) represent direct and indirect attractive forces between an electron and a hole in a NR. The last two terms correspond to repulsive forces induced by mirror charges at NR/medium dielectric boundaries. Equation (3) is spatially averaged over electron/hole wavefunctions, reducing $U(\mathbf{r}_e, \mathbf{r}_h)$ to a 1-dimensional potential. This representation is subsequently used to calculate exciton binding and self-interaction energies which, together with quantum size levels, describe the evolution of CdSe's electronic structure across dimensionality (Supplementary Note 4).

Figure 4 shows calculated transition energies for the first ($1\Sigma_{3/2}1\Sigma_e$) and second ($1\Sigma_{1/2}1\Sigma_e$) 1D-excitons as functions of NR aspect ratio. Superimposed are extracted (average) α-energies, as well as tabulated equi-diameter QD literature values[2,22,23]. $1\Sigma_{1/2}1\Sigma_e$ ($1\Sigma_{3/2}1\Sigma_e$) is predicted to be bright under parallel (perpendicular) polarized light. Consequently, we have previously assigned α in CdSe NWs to the $1\Sigma_{1/2}1\Sigma_e$ exciton[12,14,16]. In QDs, α arises from $1\Sigma_{3/2}1\Sigma_e$ ($1S_{3/2}1S_e$ in QD literature)[13,24]. The α assignment therefore shifts from $1\Sigma_{1/2}1\Sigma_e$ to $1\Sigma_{3/2}1\Sigma_e$ with decreasing aspect ratio due to a ($b/d$)-dependent transition strength. Figure 4 denotes this through $1\Sigma_{1/2}1\Sigma_e/1\Sigma_{3/2}1\Sigma_e$ curve transparencies.

Predicted transition energies are in excellent agreement with experimental NW, $b \sim 160$ nm NR and QD results. Of note is that average $b \sim 30$ nm NR α energies are higher than theoretically-derived energies. This stems from sampling slightly smaller rods within the ensemble's residual size-distribution (Supplementary Note 6). Figure 4 plots α obtained from the $b \sim 30$ nm NR ensemble spectrum relative to the single NR α-average. Approximations in the model[14] also potentially contribute to deviations between experiment and theory (Supplementary Note 7). In general though, observed experimental and theoretical trends are in qualitative and, in some cases, quantitative agreement.

## Discussion

Given that the general evolution of a semiconductor's dimensionality is described by (Fig. 4), at what point does the 1D-to-0D transition occur? Since the only aspect ratio-dependent energies contributing to a nanostructure's overall $E_g$ are its longitudinal confinement ($E_k(b)$) and electrostatic ($|U|$) energies, the 1D-to-0D evolution is characterized by the interplay between these two terms (Fig. 1). Supplementary Fig. 6 plots $E_k(b)$ and $|U|$ as functions of aspect ratio. From it, we determine where $E_k(b)$ balances $|U|$ and define this to be the critical point where a 1D-to-0D transition occurs (Supplementary Note 8). In $d \sim 6.8$ nm CdSe, this length is $b \sim 8.5$ nm, which is just above its bulk exciton Bohr radius ($a_B = 5.6$ nm)[25]. The transition point is additionally sensitive to diameter and occurs at $b \sim 6$ nm ($b \sim 11$ nm) for $d = 4$ nm ($d = 10$ nm) NRs. These findings differ from previous studies which suggest transition lengths of $b \geq 30$ nm for $d = 3–6$ nm CdSe NRs[7]. The discrepancy exists because experimental blueshifts in NR absorption spectra arise from both carrier and dielectric confinement (Supplementary Fig. 6). At large $b$, dielectric confinement is the predominant source of blueshifts to NR extinction spectra. Hence, only at lengths below $b \sim 2a_B$ (ref. 26) does carrier confinement matter, and where a crossover in nanostructure dimensionality occurs.

In summary, we directly probe the 1D-to-0D transition in CdSe using single NW/NR absorption spectroscopy. These measurements expand the limits of conventional single particle microscopies by providing the first direct absorption spectra of individual CdSe NRs. Beyond this, they clearly show how excitonic blueshifts in the linear absorption depend exquisitely on the delicate balance between confinement and dielectric

contrast/confinement interactions, the latter being fundamental interactions which govern the dimensional crossover in semiconductors.

## Methods

**Sample synthesis.** CdSe NWs ($d = 6.8 \pm 1.2$ nm) and NRs ($d = 6.7 \pm 1.1$; $b = 160 \pm 55$ nm and $d = 6.8 \pm 0.7$ nm; $b = 30.4 \pm 2.6$ nm) were made using previously established wet chemical syntheses[12,27]. A detailed description of these methods, as well as information on sample characterization can be found in Supplementary Note 2.

**NW/NR absorption spectroscopy.** Individual CdSe NWs and NRs were probed on a homebuilt system constructed around a commercial inverted microscope body (Nikon). A schematic of the experimental setup is provided in Supplementary Fig. 1. Samples were prepared by drop casting dilute NW/NR-toluene suspensions onto methanol cleaned, flamed fused silica microscope coverslips (40/20 scratch/dig; UQG Optics). The suspensions were allowed to dry with sample coverages of $\sim 1$ particle per $\sim 5 \, \mu m^2$.

Loaded coverslips were then affixed to an open-loop 3-axis piezo stage (Nanonics) coupled to a closed-loop 3-axis piezo stage (Physik Instumente) and a 2-axis mechanical stage (Semprex) for fine and coarse particle positioning, respectively. The open-loop piezo stage supplied the spatial modulation of individual particles (peak-to-peak particle displacement $\sim 360$ nm) at 750 Hz. A 2D survey ($x$–$y$, lateral) absorption map was first obtained to identify and locate single NWs/NRs. This was accomplished by scanning the sample through the focused excitation point-by-point over an $x$–$y$ grid (400 nm increments) while simultaneously detecting the corresponding absorption signal (lock-in time constant, $\tau = 30$ ms; integration time, $t_{int} = 100$ ms). These $\sim 25 \times 25 \, \mu m$ survey absorption images were then used to locate individual particles from where likely candidates were positioned in the laser focus by maximizing the signal from the lock-in amplifier. Absorption spectra were subsequently acquired by scanning the excitation wavelength ($\lambda$) through the visible (450–750, 2 nm steps) while synchronously detecting the absorption signal ($\tau = 1$ s and $t_{int} = 10$ s). Data acquisition, particle positioning, and wavelength scanning were all controlled using home-written software (C++).

Identification of individual NWs/NRs was established through a three-point vetting process. First, the absorption image was examined to ensure that no extraneous absorbers were within $\sim 3 \, \mu m$ of the particle. Next, the absorption signal was converted into $\sigma_{ext}$ and compared with expected extinction cross-sections[12,16,17]. Finally, the absorption spectrum was obtained and examined for the presence of clear transitions.

**Data availability.** The data that support the findings of this study are available from the corresponding author (M.K.) upon request.

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

## Acknowledgements

We thank P. Tongying for assistance in synthesizing nanorod samples. We also thank A. Shabaev and Al. L. Efros for fruitful discussions. M.K. acknowledges the National Science Foundation (grant no. CHE 1208091) for financial support. We also thank the Notre Dame College of Science and Departments of Chemistry/Biochemistry and Physics for financial support.

## Author contributions

M.K. and M.P.M. conceived and designed the experiments; M.P.M. and R.C. performed the experiments; M.P.M. and R.C. analysed the data; M.P.M., J.S. and R.C. developed the theoretical model under the supervision of B.J. and M.K. M.P.M., R.C., B.J. and M.K. co-wrote the paper.

## Additional information

**Competing financial interests:** The authors declare no competing financial interests.

