## [Peer review file · Nature Communications]

Reviewer #1 (Remarks to the Author):

The current manuscript NCOMMS-16-09631 titled "Dimensional crossover in semiconductor nanostructures", by Masaru Kuno¹ and co-authors, opens a wide range of possible activities in the area of optical wavelength conversion in colloidal CdSe nanostructures having 1D (wire, rod) shape. It is of a great interest to scientists because of the potential application of these nanostructures, emitting in visible wavelength range, in opto-electronics (light emitting devices and solar cells). The role of the size (diameter and length) in these 1D nanoparticles, which controls their optical bandgaps, have been previously investigated theoretically as well as experimentally. The presented manuscript proposed new idea that gives opportunity to understand how the critical NRs length at which 1D-to-0D transition occurs is governed by the aspect ratio-dependent interplay between carrier confinement and dielectric contrast/ confinement energies. The authors report on a first comprehensive, important and comparable study of dimensional crossover of the band gap in semiconductor nanostructures from one dimensional (1D) CdSe nanowires (NWs) and different length nanorods (NRs) to the same diameter quantum dots (QDs) using single NW/NR absorption spectroscopy. They also constructed a modified version of an effective mass model to explain absorption spectra of individual CdSe NWs and consider the dimensional evolution of carrier electrostatic (dielectric contrast/confinement) effects on calculated transition energies for the first and second 1D excitons as a function of NR sizes aspect ratio. I find this work important and useful as it emphasizes the significance of the understanding of the optical bandgaps nature in 1D nanostructures for their application in optical devices. The results presented in the manuscript are novel and this contribution is significant for the scientific community working on colloidal nanostructures, as well as to fundamental physics in general. I think that this paper will be interesting and useful for large community of scientists working in nanoscience and on this ground support the manuscript publication in the Nature Communications. However, the description requires some revision, which I would denote as minor:

1. The authors did not give an adequate literature review of dimensional crossover in semiconductor 1D nanostructures, which are recently investigated. For example few of them: (i) Tilchin and co-authors showed in ACS Nano 9, 7840 (2015) the evolution of 0D-to-3D dimensional crossover by monitoring the exciton size in single CdTe colloidal QD of various sizes; (ii) I. Climente and co-worker investigated strong configuration mixing due to dielectric confinement in semiconductor nanorods in Phys. Rev. B 79, 161301 (R),(2009).

2. I also think that schematic presentation of 1D-to 0D dimensional crossover, shown in Figure 1 of the main text, must be replaced by Figure S6 (a, c), which, for my opinion, described the essential idea: quantitate transition energies which occurs by the aspect ratio-dependent interplay between carrier confinement and dielectric contrast/confinement energies.

Description of the proposed manuscript is of high quality; the results are novel and interesting to the readers of Nature Communications.

In summary, I recommend this work for publication in Nature Communications after the authors address the issue listed above.

Reviewer #2 (Remarks to the Author):

This paper reports novel results on the effect of length of 1D CdSe nanowires (NWs) on the confinement of excitons. Innovative measurements of the optical absorption spectra of single NWs with varying length are reported. The experimental data have been analyzed on the basis of electronic structure calculations.

Comparison of theoretical and experimental data provides information about the factors that govern for which NW length the exciton undergoes a transition from 1D character to 0D (as in a quantum dot). The transition is shown to result from a subtle interplay between longitudinal spatial exciton confinement and effects of electrostatic energy.

The paper is written in a clear way and can be published as it is.

Author response to reviewer comments and suggestions

Reviewer comments (*italic*) followed by our responses (**boldface**) are supplied below.

Reviewer 1

The current manuscript NCOMMS-16-09631 titled "Dimensional crossover in semiconductor nanostructures", by Masaru Kuno1 and co-authors, opens a wide range of possible activities in the area of optical wavelength conversion in colloidal CdSe nanostructures having 1D (wire, rod) shape. It is of a great interest to scientists because of the potential application of these nanostructures, emitting in visible wavelength range, in opto-electronics (light emitting devices and solar cells). The role of the size (diameter and length) in these 1D nanoparticles, which controls their optical bandgaps, have been previously investigated theoretically as well as experimentally. The presented manuscript proposed new idea that gives opportunity to understand how the critical NRs length at which 1D-to-0D transition occurs is governed by the aspect ratio-dependent interplay between carrier confinement and dielectric contrast/confinement energies.

The authors report on a first comprehensive, important and comparable study of dimensional crossover of the band gap in semiconductor nanostructures from one dimensional (1D) CdSe nanowires (NWs) and different length nanorods (NRs) to the same diameter quantum dots (QDs) using single NW/NR absorption spectroscopy. They also constructed a modified version of an effective mass model to explain absorption spectra of individual CdSe NWs and consider the dimensional evolution of carrier electrostatic (dielectric contrast/confinement) effects on calculated transition energies for the first and second 1D excitons as a function of NR sizes aspect ratio.

I find this work important and useful as it emphasizes the significance of the understanding of the optical bandgaps nature in 1D nanostructures for their application in optical devices. The results presented in the manuscript are novel and this contribution is significant for the scientific community working on colloidal nanostructures, as well as to fundamental physics in general. I think that this paper will be interesting and useful for large community of scientists working in nanoscience and on this ground support the manuscript publication in the Nature Communications. However, the description requires some revision, which I would denote as minor:

1. The authors did not give an adequate literature review of dimensional crossover in semiconductor 1D nanostructures, which are recently investigated. For example few of them: (i) Tilchin and co-authors showed in ACS Nano 9, 7840 (2015) the evolution of 0D-to-3D dimensional crossover by monitoring the exciton size in single CdTe colloidal QD of various sizes; (ii) I. Climente and co-worker investigated strong configuration mixing due to dielectric confinement in semiconductor nanorods in Phys. Rev. B 79, 161301 (R) (2009).

We thank the reviewer bringing these publications to our attention. We now cite them within the manuscript. In particular, we cite [Tilchen, et al, ACS Nano (2015)] on page one of the manuscript (ref. 3), and [Climente, et al, PRB (2009)] on page seven of the manuscript (ref. 21).

2. I also think that schematic presentation of 1D-to 0D dimensional crossover, shown in Figure 1 of the main text, must be replaced by Figure S6 (a, c), which, for my opinion, described the

essential idea: quantitate transition energies which occurs by the aspect ratio-dependent interplay between carrier confinement and dielectric contrast/confinement energies.

We have now merged Figure S6 into Figure 1 by replacing the schematic representation of carrier confinement [$E_k(b)$ and $|U|$ versus aspect ratio] with actual data from the theoretical model (from Figure S6). In particular, Figure 1 has been changed from

to

We believe that this new figure better captures the essential idea of the manuscript, while also retaining its introductory nature.

Reviewer 2

This paper reports novel results on the effect of length of 1D CdSe nanowires (NWs) on the confinement of excitons. Innovative measurements of the optical absorption spectra of single NWs with varying length are reported. The experimental data have been analyzed on the basis of electronic structure calculations.

Comparison of theoretical and experimental data provides information about the factors that govern for which NW length the exciton undergoes a transition from 1D character to 0D (as in a quantum dot). The transition is shown to result from a subtle interplay between longitudinal spatial exciton confinement and effects of electrostatic energy. The paper is written in a clear way and can be published as it is.

We thank the reviewer for the positive feedback.

Editorial

I would like to highlight that the Results section must be split into subheaded sections; the subheadings should be no longer than 60 characters including spaces. Subheadings should contain no punctuation.

We have added suitable subheadings to the Results section in accordance with the Nature Communications format.